# The Impact of the ‘Planning Health in School’ Programme on a Pair of Twins with Obesity

**DOI:** 10.3390/children9121866

**Published:** 2022-11-30

**Authors:** Margarida Vieira, Graça S. Carvalho

**Affiliations:** Research Centre on Child Studies, University of Minho, 4710-057 Braga, Portugal

**Keywords:** childhood obesity, health promotion, school-based intervention, twins study, obesity prevention, children’s health

## Abstract

This case study describes the impact of the ‘Planning Health in School’ programme (PHS-pro) on the nutritional status and lifestyle behaviours of two twins with obesity. As part of a larger research project involving 449 adolescents in grade-6, PHS-pro aims at preventing obesity and guiding children towards healthy behaviours. Twins were evaluated for anthropometric measurements—height, weight, body mass index (BMI), waist circumference (WC), and lifestyle behaviours before (baseline) and after (8 months) PHS-pro and at a follow-up (one-year later). At the baseline, both twins were obese according to the international cut-off points of Cole. After PHS-pro, improvements in anthropometric parameters were found: the boy decreased his BMI by 10% and lost 9.0 cm in WC, while remaining obese; the girl decreased her BMI by 8% and lost 8.7 cm in WC, changing to the overweight category. At the follow-up, a slight increase in the anthropometric parameters was found in both twins; however, they did not return to the baseline values. The programme successfully promoted positive changes in behaviours and improved nutritional status, showing the long-term effects of the PHS-pro. Although it is a school-based intervention to prevent obesity, the PHS-pro is helpful in weight reduction even in children already with obesity.

## 1. Introduction

Obesity is a strong negative burden on children’s individual health, on their families, and on the overall public health care system of a nation. Effective interventions to reduce and solve obesity are required, and prevention is the most effective answer [1].

Since 1980, the prevalence of obesity has doubled worldwide and is currently a problem crossing cultures and countries. Among children, the numbers show a similar proportion [2]. A systematic analysis of overweight and obesity in children has shown a prevalence above the 20%, being observed in both developed and developing countries [3]. Childhood obesity significantly increases the early risk of chronic diseases linked to an unhealthy diet and physical inactivity, with disastrous economic predictions [4,5]. Costs to treat obesity and consequent diseases for the entire population are unbearably high, requiring urgent motivation and guidance to change people’s behaviours, in order to undergo prevention through healthy eating and active lifestyles. Several studies have confirmed the growing trends of very low consumption rates of fruit and vegetables, with high consumption rates of high-fat and energy-dense foods, added sugar food, and sugary drinks, as well as the lack of breakfast among adolescents [6,7,8], which worsens the current situation.

A meta-analysis of health promotion interventions puts forward that the most effective interventions should increase people’s awareness about their daily habits and involves them changing their behaviour in order to obtain positive results and to have a higher chance of achieving public health goals when compared to passive interventions that only provide information on obesity risks [9].

Taking all this into consideration, a school-based intervention called the “Planning Health in School” programme (PHS-pro), grounded on the Transtheoretical Model (TTM) and the five stages of change developed by Prochaska and colleagues [10], was designed and implemented to prevent children’s obesity.

The TTM offers an understanding of how people develop a new behaviour and improve skills and self-efficacy, by assessing the stage of readiness for change. Based on this, the PHS-pro was developed so that children would progress through the five stages identified by Prochaska and colleagues: (i) precontemplation—no intention to change behaviour or take action in the near future, unawareness of their problems; (ii) contemplation— aware of unhealthy habits, interest and intention to change in the next 6 months; (iii) preparation—intending to take action in the next 30 days; (iv) action—making changes and visible modifications on specific behaviour targets along 6 months; (v) maintenance— working to continue and consolidate healthy habits [10]. Thus, the PHS-pro and its educational contents followed the five-stages of TTM readiness, for moving children from inaction to action or to maintenance, in order to successfully change behaviour [10].

Additionally, the application of the participatory methodology was used to engage children to meet their needs, including them as active participants in their own change process [11]. The PHS-pro aimed to improve children’s eating behaviour and active living by implementing a nutritional education programme shaped as eight learning modules and delivered monthly over an academic year [12].

The aim of this case study is to describe the impact of the PHS-pro on the nutritional status and lifestyle behaviours of two twins with obesity, a boy and a girl of 11 years old, over three specific evaluation moments: before the PHS-pro (baseline), at the end of the PHS-pro (8 months), and one year after the end of the PHS-pro (follow-up).

## 2. Materials and Methods

The PHS-pro is a school-based intervention programme implemented for grade 6 children, aged 10 to 14, over a full academic year (2011/2012) at the four existing elementary schools of a municipality (Trofa) integrated into the second-largest metropolitan area of Portugal—Porto [13]. Informed consent was requested among all the children and their parents before the study. The Scientific Council of the Institute of Education of the University of Minho granted ethical approval, and ethical permission was obtained from the School Pedagogical Board. Children’s participation rates were high (>99%).

A non-randomised control group pretest–posttest trial was conducted at the four existing elementary schools of the municipality to assess the effectiveness of the programme. The final sample, composed of 449 children, was allocated into two groups: the intervention group with 219 children (115 boys and 104 girls) of one school and the control group with 230 children (111 boys and 119 girls) of the other three schools [13]. The mean age was 11.2 ± 0.6.

During the data organization process, a pair of twins with obesity was detected in the intervention group: a boy and a girl of 11 years old. Therefore, it was determined to continue the assessment of the twins over a longer period with a follow-up after the end of the programme.

In order to improve knowledge, skills, and behaviours regarding healthy eating and active living, the PHS-pro integrated a set of eight learning modules, delivered once a month over the school year, and monitored through food records [14].

The eight learning modules focused on four main goals for changing behaviour on a daily basis: the adequate consumption of fruit and vegetables—five servings/day; decreasing high-sugar food and beverage intake (to 10% of free sugar of total daily energy); decreasing high-fat and energy-dense food consumption; and 1 h of physical activity and no more than 2 h spent watch TV. Each learning module (LM) addressed a different topic with a specific change in behaviour and the sequence’s themes were as follows: LM1—“10 steps to be healthier”—key message: selecting one or two steps between the 10 to improve as a behaviour change goal; LM2—“Water & milk help you to grow up” –key message: adopting water or milk instead of soft drinks; LM3—“Training every day to be healthier”—key message: thinking of a strategy to change their families’ behaviour to have a healthier life at home; LM4—“3 fruits a day, how much good it does?”—key message: setting a specific daily fruit serving goal to be accomplished for the following weeks; LM5- “Fruit and vegetables are essential to life”—key message: make children understand what their vegetable preferences are in order to begin eating their favourite vegetables to reach a healthy goal; LM6—“Start on moving!”- key message: to begin practicing a sport or other forms of physical activity; LM7—“The best snacks”—key message: implementing snacks rich in essential nutrients; LM8—“Final game: who has learned about everything?” –key message: programme overview and children’s self-evaluation [12].

For the standard evaluation process, two assessments were conducted before the programme (baseline) and after 8 months (post PHS-pro), but for this study of the twins, a third assessment was carried out 1 year after the PHS-pro finished (follow-up). The following two methodological procedures were used in the standard evaluations:(1)Anthropometric assessment to collect body composition measures: weight, height, and waist circumference and calculation of body mass index (BMI). All measures were taken according to standardised international procedures [15]. International cut-off points of Cole and collaborators [16] by age and gender were used for classifying the prevalence of overweight and obesity. For waist circumference (WC), the reference values of McCarthy [17] by age and gender were adopted to correlate the percentile curves with the deposition of intra-abdominal fat mass, being percentile 90 (≥ P90), the cut-off point used to identify excessive central fatness [18].(2)The application of a self-report questionnaire composed of three distinct sections: a 58-item food frequency questionnaire (FFQ) to evaluate eating behaviour [14]; 8 questions on healthy eating and basic food knowledge; and 6 questions on sports, physical activities, and leisure habits such as television, videogames, and computers (Appendix A
Table A1).

Moreover, at the end of the PHS-pro (post-PHS-pro), another questionnaire, supported by semi-structured interviews, was applied to obtain children’s insights regarding the potential changes observed in their eating and lifestyle behaviours during the programme. Children were specifically asked if they recognized any changes in their own eating behaviours and lifestyle over the intervention through a yes-no question, followed by an open space to report the identified changes. The three questions used were as follows: (1) “Over these last months while the learning modules were being developed, did you change any of your eating behaviours or physical activity? If yes, please list what do you do now, that you didn’t do before”; (2) “Were the learning modules helpful for you to change behaviours? If yes, which were most helpful”; (3) “Do you have suggestions or ideas to propose, so that the programme can help more children to improve their diet and other lifestyle behaviours?”

Throughout the PHS-pro intervention, the primary goal was to monitor eating behaviour and to evaluate children’s progress after each learning module. In this way, the 3-day food record was chosen as the principal tool for measuring children’s food consumption and food choices, allowing them to report all the food and beverages they consumed for 3 days in high levels of detail [19]. To achieve this, a food record form was built to be adapted to the features of the research. The 3-day food record was applied before the PHS-pro intervention and after each learning module over the PHS-pro period implementation, making a total of seven food records [12]. Like all the other children, both twins delivered their food records. One year after the end of the PHS-pro, at the follow-up, a 3-day food record was applied to the twins using the same methodological procedures, for measuring their food consumption and food choices again.

## 3. Results

Nutritional literacy is a crucial requirement in the process of improving food choices and eating behaviours. Therefore, healthy eating and basic food knowledge were evaluated by answers to eight questions during the two evaluation points of the programme (baseline and post PHS-pro). At the baseline, the twins showed different levels of knowledge: the boy answered all the questions correctly, getting a score of 100% at both evaluations; the girl improved from a score of 37.5% at the baseline to 62.5% at the post PHS-pro.

### 3.1. Changes in the Twins’ Nutritional Statuses

The twins’ anthropometric data and their progression along the three evaluations are separately outlined in Figure 1 and Figure 2.

At the baseline, both twins were considered obese according to the international cut-off points of Cole [16]: the boy had a BMI of 37.47 kg/m^2^ (the cut-off point value for obesity in boys at 11 y-old is 25.10) and the girl 26.46 kg/m^2^ (the cut-off point for obesity in girls at 11 y-old is 25.42). The WC value recorded for the boy was 117.1 cm and for the girl it was 91.3 cm, placing them on P95 in the reference values of McCarthy [17].

After the eight months of the programme implementation (post PHS-pro), these two anthropometric measures improved: the boy decreased both his BMI to 33.67 (10.0%) and his WC to 108.2 cm (7.6% less) but remained in the category of obesity (see Figure 1); similarly, the girl decreased her BMI to 24.34 (8.0%) and WC to 82.5 cm (9.6% less), leading to a positive change from the classification of obesity to overweight (see Figure 2).

During the PHS-pro both children naturally grew in height (159.53 cm to 163.40 cm for the boy and 146.33 cm to 150.87 cm for the girl), yet they lost weight (95.4 kg to 89.9 kg for the boy and 56.6 kg to 55.4 kg for the girl), with the loss occurring more markedly in the boy, who lost 5.5 kg (see Figure 1 and Figure 2).

One year after the PHS-pro finished, at the follow-up evaluation, both twins showed changes in all anthropometric measures: the boy increased both his BMI to 34.13 (1.4% increase) and his WC to 108.6 (0.36% more), continuing in the obesity category (the obesity cut-off point value for 12.5 y-old boys is 26.43), although he did not return to his initial values of BMI and WC recorded at the baseline (37.5 and 117.1 cm, respectively). The girl kept the category of overweight with a BMI of 26.54 (the obesity cut-off point for 12.5 y-old girls is 27.24), however over the 12 month period, she showed a 2-point increase in BMI, and 6.4 cm more in the WC, reaching 88.9 cm. Nevertheless, even with this increase, the girl was no longer obese, as she was at the baseline (26.46 and 91.3 cm, respectively).

### 3.2. Changes in the Twins’ Eating and Lifestyles Behaviours

Based on data collected through the FFQ, the eating behaviour improved in both twins. The boy’s behaviour changes were found in the decrease of the consumption frequency of high fat and high added-sugar foods per week, while in the girl, the major changes took place in the increase of the consumption frequency of fruit and dairy products. Specifically, the boy reported: eating fewer smoked meats such as smoked ham and salami; decreasing the frequency of cookies and biscuits consumption from “2 to 3 times per week” to “never or less than one time per week”; and reducing the consumption of patisserie products (croissants, donuts, pastries and cakes), processed sausages, French fries, and sauces (mayonnaise and ketchup) from “one time per week” to “never or less than one time per week”. He also stopped eating daily breakfast cereals and bread slices as a complement to the main dish at lunch and dinner.

The girl reported several improvements: the increase in milk intake by drinking “4 to 6 times per week” rather than “never or 1 time per week” and yoghurt with a frequency of “4 to 6 times per week” instead of “2 to 3 times per week”; greater fruit consumption from “2 to 3 times per week” to “4 to 6 times per week”; and a positive breakfast change from not eating breakfast to having it “2 to 3 times per week”.

Changes in the twins’ eating behaviour through 3-day food records for vegetable soup, fruit, and salad consumptions are presented in Table 1. Out of the seven food records applied over the PHS-pro to monitor it, each twin delivered four food records properly filled in. Portions of vegetable soup, salad, and fruit recorded on the food records were summed and calculated as the average portions.

Thus, during the PHS-pro, the boy recorded an increase in intake of vegetable soup (2 to 3) and salad intake (0 to 1) but a reduction in fruit (2 to 1), whereas the girl only increased vegetable soup intake (2 to 5) and decreased both fruit (3 to 0) and salad (4 to 3) intake (see Table 1). The 3-day food record applied one year later, at the follow-up, reported a decrease in vegetable soup intake for the boy (3 to 0), which might suggest the loss of the new daily behaviour of eating vegetable soup established over the PHS-pro. On the other hand, the girl increased the vegetable soup (5 to 6) but decreased the salad intake (3 to 2).

Changes between the baseline and post the PHS-pro were detected in the sports, physical activities, and leisure behaviours in six questions of the self-reported questionnaire: the boy reported that he enrolled in a futsal team, practising between 4 to 6 h per week, whereas the girl said she did not practice any extracurricular sport and remained sedentary (see Table 2). Accordingly, the boy reported spending less time watching TV per day, from “2 to 3 h” to “30 min to 1 h”, but the girl continued unchanged spending “30 min to 1 h” watching TV.

One year later, at the follow-up, the boy reported quitting the extracurricular practice of futsal, and only kept the physical education classes at school. The girl continued unchanged, only attending the two classes of physical education of 135 min per week (see Table 2).

### 3.3. Changes in Twins’ Perceptions

The twins’ perceptions were assessed with a questionnaire and supported by semi-structured interviews to understand whether they changed their eating behaviours and lifestyles during the PHS-pro. The boy defined his participation with the following statement: “I changed my way of thinking” and perceived the need to make efforts to eat vegetable soup and salad daily because he recognised that he liked eating this kind of food, even though he was not used to it. The twin sister claimed that, during the PHS-pro, she began to give more attention to what she ate, and included daily vegetable soup, salads, and fish-based meals in her diet. Additionally, she said that she would like to do some physical exercise and participate in sports activities, and that she felt more confident in changing her weight through healthier lifestyle behaviours. Nonetheless, she never joined any extracurricular sports or began any kind of physical activity, except the two physical education classes at school.

## 4. Discussion

As far as we know, this study is the first to track a pair of adolescent twins with obesity participating in a school-based intervention to promote healthy eating and physical activity. Since both twins had a beneficial evolution over the intervention (from the baseline to post-PHS-pro), we decided to make a one-year follow-up assessment to find out whether the programme would have a long-term effect. Although the results of this study of twins cannot be applied generally, it may be of interest in the context of lifestyle-related illnesses and diseases, and their prevention, since the heterozygous twins share a common gene pool and live in the same socioeconomic environment with a family that provides the same resources and ways of living [20].

In general, the nutritional status variables analysed in this study showed that both twins had a beneficial evolution with the PHS-pro intervention, which persisted up to the one-year follow-up evaluation, confirming the value of the school-based interventions. Indeed, both twins reduced their BMI values and WC measures and did not return to their baseline values. Both twins’ results were mainly influenced by changes in food choices and eating behaviours. In addition, the boy also began to do sports activities.

According to this, the PHS-pro did help to develop the motivation and skills to choose healthy behaviours.

It is worth noting that, although the girl remained in the overweight category and was no longer obese, over a 12-month period her BMI recorded a 2-point increase and 6.4 cm more in the WC. Similarly, the boy also had a regression, which was slower than the girls, with a 0.5-point increase in BMI and a 0.4 cm increase in WC. These variations can be explained by the fact that, after the PHS-pro, the boy quit the extracurricular physical activity he engaged in during the PHS-pro. More time spent in physical activity, from 4 to 6 h per week engaged in futsal, had led him to meet the recommendations on physical activity, and subsequently less time was spent on sedentary tasks, such as watching TV and gaming.

Therefore, in order to keep the changes acquired by the new behaviours and maintaining the involvement of the children, it seems crucial to maintain the programme in order to properly sustain the results achieved.

The twins’ lifestyle behaviours, in particular their eating behaviour, were monitored, revealing an imbalanced dietary pattern at the baseline: a low consumption of fruit and vegetables, pulses, and dairy products, which are well-known for being rich in vitamins, dietary fibres, and minerals such as iron and calcium, supplemented with the regular consumption of foods high in calories, and high sugar beverages. In addition to this, the physical inactivity of both twins also contributed to their obesity status.

Considering that each food record corresponded to three days of records, from the start of the programme (baseline), this pair of twins reported a very low consumption of fruit and vegetables. Despite the overall improvement in the intake of these foods during the programme, the results clearly show that these twins continued to have a poor consumption rate of fruits and vegetables, remaining far behind daily recommendations and nutritional requirements, which comprises an adequate supply of vitamins, minerals, and dietary fibres, that are considered essential nutrients in their age group for healthy physical and intellectual development [21].

As the PHS-pro was implemented, there were improvements in the twins’ attitudes and behaviours, indicating their awareness and a great concern to adopt healthy eating behaviours and desire to be more physically active. These new perceptions and behaviours have provided visible changes that impacted the nutritional statuses of the twins by the end of the programme.

As children and adolescents become increasingly more independent to make their own choices, the implementation of programmes like this one, which propose to create and support healthy eating, as well as to change behaviours that can help to stop the rise of obesity rates, must be kept active for longer, without gaps or great breaks in time.

Given that this is a case report, limitations should be acknowledged. First, this study is limited as the small sample size is only restricted to one pair of twins living in the same home and school conditions, hence no causal inferences can be made, and caution is needed in the interpretation of results. Second, the aim of this study was to focus on the evolution of this pair of 11 year-old twins over the programme, but as they were able to respond themselves and were auto-evaluated, the study could be limited by the reporting bias caused by the self- reporting questionnaires and the children’s own perceptions. However, the trained investigator throughout the programme can ensure the validity of the anthropometric measures.

Moreover, the results showed that it is possible to follow children with obesity during childhood in a school setting, to guide their growth, and to encourage them to change behaviours, in order to get positive results step-by-step in a consistent and sustained way.

## 5. Conclusions

This case study of twins showed that the participation in the PHS-pro has successfully changed their attitudes and behaviours towards a healthy diet and lifestyle, which enabled a positive evolution in their nutritional status, up to at least one year. Additionally, it suggests that, although the PHS-pro is a school-based intervention targeted to prevent overweight and obesity, it is also helpful in weight reduction, even when children are already obese.

## Figures and Tables

**Figure 1 children-09-01866-f001:**
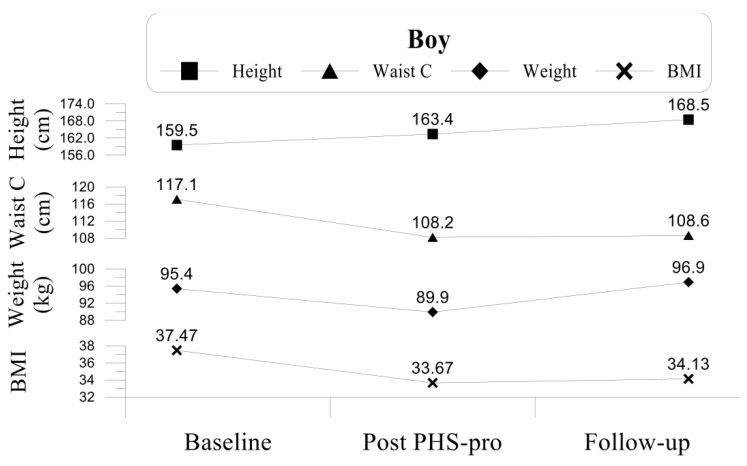
Anthropometric data of the boy.

**Figure 2 children-09-01866-f002:**
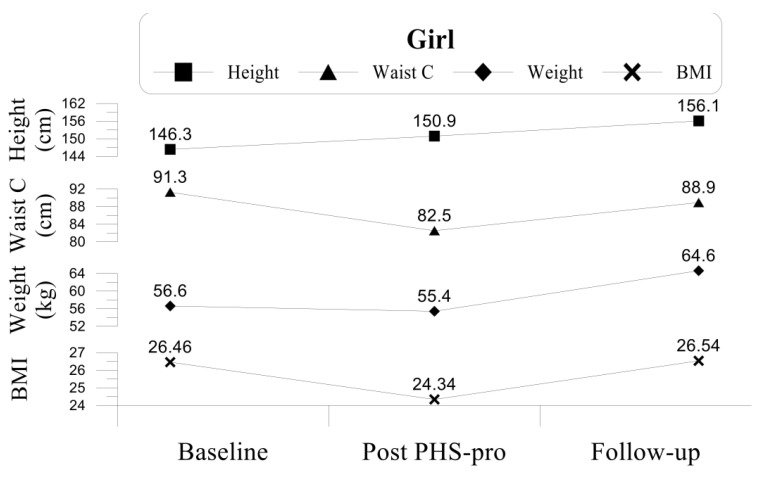
Anthropometric data of the girl.

**Table 1 children-09-01866-t001:** Consumption of vegetable soup, fruit, and salad in the 3-day food record.

Twins	Baseline (3-Day Food Record)	PHS-Pro (Average of 4 Food Records)	Follow-Up (3-Day Food Record)
Boy	Vegetable soup – 2	Vegetable soup – 3	Vegetable soup – 0
	Fruit – 2	Fruit – 1	Fruit – 2
	Salad – 0	Salad – 1	Salad – 1
	Total = 4 portions (1.3/day)	Total = 5 portions (1.7/day)	Total = 3 portions (1/day)
Girl	Vegetable soup – 2	Vegetable soup – 5	Vegetable soup – 6
	Fruit – 3	Fruit – 0	Fruit – 0
	Salad – 4	Salad – 3	Salad – 2
	Total = 9 portions (3/day)	Total = 8 portions (2.7/day)	Total = 8 portions (2.7/day)

**Table 2 children-09-01866-t002:** Physical Activity time of both twin over the three evaluations.

Twins	Baseline	Post PHS-Pro	Follow-Up
Boy	135 min/week *	135 min/week *	135 min/week *
		Extracurricular sport	Quit the futsal
		Futsal: 4 to 6 h	
Girl	135 min/week *	135 min/week *	135 min/week *
		No extracurricular sport	

* related to two classes of physical education at school.

## Data Availability

The data presented in this study are available on request from the corresponding author. The data are not publicly available due to data privacy of the participants.

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
