# Peer review of "The Impact of the ‘Planning Health in School’ Programme on a Pair of Twins with Obesity"

_children, 2022, doi:10.3390/children9121866_

Round 1

Reviewer 1 Report

1.       The Transtheoretical Model should be explained in detail (eg what are the stages, what interventions are recommended at each stage?) (line 47)

2.       There is confusion in the study design. Groups and design should be expressed more clearly for readers.

449 adolescents

intervention group and 230 children (111 boys 68 and 119 girls) allocated in the control group. (line 68)

a pair of twins (a boy and a girl of 11 years 74 old) with obesity was found in the intervention group (line 74).

3. The questionnaire used in the semi-structured interview should be detailed (line 92).

4. It is mentioned that the program encourages physical activity. But there is no information on how it promotes physical activity (line 206).

5. The limitations of the study were not defined.

Author Response

We sincerely thank the reviewers for careful reading of our manuscript and the constructive comments and suggestions, which allowed helping revising it. Our response follows the reviewer comments.

Review 1

  1. The Transtheoretical Model should be explained in detail (eg what are the stages, what interventions are recommended at each stage?) (line 47)

Response: We would like to thank you for your valuable comments concerning our manuscript and we have revised accordingly (marked in underline-red). Following the suggestions that the Transtheoretical Model should be explained in detail, it is now described in the Introduction section.

  1. There is confusion in the study design. Groups and design should be expressed more clearly for readers.

449 adolescents

intervention group and 230 children (111 boys 68 and 119 girls) allocated in the control group. (line 68)

a pair of twins (a boy and a girl of 11 years 74 old) with obesity was found in the intervention group (line 74).

Response: Regarding “the groups and design should be expressed more clearly”, we have added more information to clarify, and we also included the reference of the study that fully detailed the study design of the programme’s effectiveness, for the interested reader.

  1. The questionnaire used in the semi-structured interview should be detailed (line 92).

Response: As suggested by the reviewer to detail the questionnaire used in the semi-structured interview, it is now described after line 92.

  1. It is mentioned that the program encourages physical activity. But there is no information on how it promotes physical activity (line 206).

Response: With regard to comment of ‘there is no information on how the programme promotes physical activity’, we add information about the educational contents of the programme, explaining the sequence of the learning modules developed during the PHS-pro and its key messages in the 5th paragraph of the ‘Materials and Methods’ section. The learning module 6 (LM6) developed the theme “Start on moving!” with the key message of beginning the practice of a sport or other forms of physical activity, in order to increase levels of physical activity among children participants. Also, we add the reference for the interested reader can refer for more information.

Reference: Vieira,M.;Carvalho,G.S. Children Learn, Children Do! Results of the “Planning Health in School”, a Behavioural Change Programme. Int. J. Environ. Res. Public Health 2021, 18, 9872. https://doi.org/10.3390/ ijerph18189872

  1. The limitations of the study were not defined.

Response: Thank you for your helpful comment, which allow us to improve the paper.

We added a new paragraph to the ‘Discussion’ section to explain the limitations of the case study.

Reviewer 2 Report

I would like to congratulate the authors for this original research and for continuing the follow-up of the twins over a longer period than the classical duration of the PHS-pro program. 

I have some comments in order to improve the paper.

In the materials and methods section, it would be interesting to detail the contents of the 8 modules, perhaps in the form of a table summarizing the themes covered and the key messages delivered at the end of each module. In the same idea, it would be necessary to add as supplemental data the  8 questions on healthy eating and basic food knowledge and the 6 questions on sports, physical activities and leisure habits such as television, videogames and computer.

In the results section, comments about follow-up at 12 months after the end of the program should be moderated. In this sense, Line 132 the term "slight" should be modified and the 2-point increase in BMI in girls should be clearly stated and discussed in the discussion section.

For physical activity, follow-up data should be added and ideally presented in a table format to see the evolution. This addition, which may highlight a decrease in active time and could explain the increase in BMI.

Line 186 : It is necessary to remove an "in". 

In the discussion section, a paragraph should be added about the need for recalls after the program in order to maintain behavioral changes 

Author Response

We sincerely thank the reviewers for careful reading of our manuscript and the constructive comments and suggestions, which allowed helping revising it. Our response follows the reviewer comments.

Review 2

I would like to congratulate the authors for this original research and for continuing the follow-up of the twins over a longer period than the classical duration of the PHS-pro program. 

I have some comments in order to improve the paper.

Response: Thank you, your comments were extremely helpful and we have revised accordingly (marked in underline-red).
We have added more explanations, which we hope meet with your approval. We answer your comments in details in the following texts.

In the materials and methods section, it would be interesting to detail the contents of the 8 modules, perhaps in the form of a table summarizing the themes covered and the key messages delivered at the end of each module.

Response: Thank you for this excellent observation. We strongly agree with the comments.
As suggested by the reviewer to detail the content of the eight learning modules in the ‘Material and Methods’ section, we have included the themes and the key messages, however since these information is available in a fully detailed table and explained in other paper, we add the reference for the interested reader can refer.

Reference: Vieira,M.;Carvalho,G.S. Children Learn, Children Do! Results of the “Planning Health in School”, a Behavioural Change Programme. Int. J. Environ. Res. Public Health 2021, 18, 9872. https://doi.org/10.3390/ ijerph18189872

In the same idea, it would be necessary to add as supplemental data the  8 questions on healthy eating and basic food knowledge and the 6 questions on sports, physical activities and leisure habits such as television, videogames and computer.

Response: The 8 questions on healthy eating and basic food knowledge and the 6 questions on sports, physical activities and leisure habits are now available in a appendix table.

In the results section, comments about follow-up at 12 months after the end of the program should be moderated. In this sense, Line 132 the term "slight" should be modified and the 2-point increase in BMI in girls should be clearly stated and discussed in the discussion section.

Response: Thank you again for your excellent observation. The term “slight” was removed and the paragraph regarding the follow-up (12 months after PHS-pro) was rephrased. We still want to add that, while we are reviewing all the numbers we identified an error in the transcription of the numbers (at 8 months) into the text and we corrected that.

For physical activity, follow-up data should be added and ideally presented in a table format to see the evolution. This addition, which may highlight a decrease in active time and could explain the increase in BMI.

Response: As suggested by the reviewer it was added a new table to present the evolution regarding the physical activity of both twins, and it was also added a sentence to describe it. This help to show more clearly that the decrease of time spent in futsal might contribute for the increase of the BMI of the boy. Yet, the girl despite she told us that she would want to do more activity; she never began anything, besides the two physical educations classes at school.

Regarding the decrease in active time that could explain the increase in BMI, it was added an explanation in the ‘Discussion’ section.

Line 186 : It is necessary to remove an "in". 
Response: it is removed.

In the discussion section, a paragraph should be added about the need for recalls after the program in order to maintain behavioral changes 

Response: Following the suggestions, a paragraph was added regarding the need of recalls after the programme.

Round 2

Reviewer 1 Report

1.     Reference should be added to the added part of the Transtheoretical Model.

Author Response

Response: Reference was in two points of the text. We added one more time at the final of the paragraph. The three points are now marked in yellow-highlighted.